



# Jet stream variability in a polar warming scenario - a laboratory perspective

Costanza Rodda[1,2], Uwe Harlander[1], and Miklos Vincze[3]

[1]Brandenburg University of Technology Cottbus–Senftenberg, Department of Aerodynamics and Fluid Mechanics, Cottbus, D-03046, Germany
[2]Department of Civil and Environmental Engineering, Imperial College London, London SW7 2AZ, England, United Kingdom
[3]von Kańmán Laboratory of Environmental Flows; Eötvös Loránd University, Budapest H-1117, Hungary/ MTA-ELTE Theoretical Physics Research Group; Eötvös Loránd University, Budapest H-1117, Hungary

**Correspondence:** Costanza Rodda (c.rodda@imperial.ac.uk)

**Abstract.**

We report on a set of laboratory experiments to investigate the effect of polar warming on the mid-latitude jet stream. Our results show that a progressive decrease of the meridional temperature difference slows down the eastward propagation of the jet stream and complexifies its structure. Temperature variability decreases in relation to the laboratory 'Arctic warming' only at locations representing the Earth's polar and mid-latitudes, which are influenced by the jet stream, whilst such trend reverses in the equatorial region south of the simulated subtropical jet. The reduced variability results in narrower temperature distributions and hence milder extreme events. However, our experiments also show that the frequency of such events increases at polar and mid-latitudes with decreased meridional temperature difference, whilst it decreases towards the equators. Despite missing land-sea contrast in the laboratory model, we find qualitatively similar trends of temperature variability and extreme events in the experimental data and the National Centers for Environmental Prediction (NCEP) reanalysis data.

## 1 Introduction

Since 1980, the polar regions have been warming approximately twice as fast as the mid-latitudes in the Northern Hemisphere, a phenomenon known as Arctic warming amplification. Model simulations (the 40 member CCSM3 ensemble) imply that this trend will continue in the future due to a robust global warming signal (Wallace et al., 2014). The signal-to-noise ratio is relatively small at higher latitudes and more significant in the tropics. Despite its well-established presence, there is still a debate about the leading causes of Arctic warming amplification. Different models suggest that sea-ice loss, lapse-rate feedback, or increasing downwelling radiation at the surface could be the main contributor (Stuecker et al., 2018).

Regardless of the causes, it is not clear whether this faster Arctic warming impacts the large-scale circulation and, if so, the effects of such changes on extreme events. Francis and Vavrus (2015) found that the Arctic amplification enlarges the North-South meandering of the mid-latitude jet-stream and causes a slow down in the eastward progression of Rossby waves. These waves are closely connected to the spatio-temporal distribution of extreme weather events. Cold outbreaks, for instance,



occur when the crests of Rossby waves penetrate lower latitudes, whilst heat waves may develop at their troughs. A slower progression of Rossby waves impacts extreme weather events—such as heatwaves, heavy downpours, and hurricanes—by increasing their duration.

Other studies have reported a reduction in the mid-latitude temperature variability, suggesting that the previously observed increase in meandering has reversed its course and questioning whether it is caused by internal variability or arctic amplification (Blackport and Screen, 2020; Dai and Deng, 2021). A lowered variability of the future mid-latitude weather would lead to less extreme temperatures.

The two major wave features that have been connected to extreme events and Arctic warming are quasi-resonant amplifica-
tion of Rossby waves trapped in a waveguide and blocking (Petoukhov et al., 2013), a persistent pressure anomaly that prevents the usual zonal propagation of atmospheric perturbations (Benzi et al., 1986). The latter is connected to Arctic warming, assuming that a reduced temperature gradient leads to a slow down of the background westerlies and the eastward progressing Rossby waves. Hence, blocking and extreme events become more likely. For example, Screen and Simmonds (2014) report a higher frequency of extreme events in coincidence with high-amplitude planetary waves that seem to favour the occurrence of
extreme weather. Other studies, however, indicate a decrease in blocks due to the decreasing the mid-latitude to pole temperature difference (Hassanzadeh et al., 2014).

These contrasting results point toward the need for more studies to help resolve the matter of Arctic amplification consequences on the mid-latitude weather. The complex dynamics of the atmosphere and the multiple influences and feedbacks, however, pose severe difficulties in finding a final answer (Overland et al., 2016). The analysis of historical changes and fully
coupled model simulations are particularly challenging by the fact that many of these phenomena are correlated and, therefore, difficult to distinguished through regression or correlation analysis (Dai and Song, 2020).

Laboratory experiments enable us to isolate the two key elements of mid-latitude variability from any other feedback process. Such isolation is fundamental in understanding the dynamical cause-effect connection. Furthermore, laboratory experiments are repeatable and can simulate very long time series, providing a statistically significant data set. Hence, we propose an
experimental study to complement the widely used observational data. The differentially heated rotating annulus (Vincze et al., 2015), which has been widely used to model the atmospheric jet stream (Hide, 1958) and nonlinear interactions between Rossby wave trains and the mean flow (Read et al., 2014), is ideal for such investigation.

The combined effect of the radial temperature difference produced by two thermal baths, analogous to the pole-to-equator temperature gradient, and the rotation of the tank, analogous to the planetary rotation, reproduce the fundamental physical
forcing of the atmospheric large-scale overturning circulation and thermal Rossby wave generation. Under certain combinations of temperature difference and rotation, the baroclinic instability can develop in the middle gap giving rise to baroclinic waves. These waves developing in the annulus are analogous to the atmospheric Rossby waves, which shape the meandering of the mid-latitude jet stream.

Over many years, these laboratory experiments have played a prominent role in geophysical fluid dynamics and climate
studies. In the review article by Vincze and Jánosi (2016), several examples are given, such as the investigation of asymmetries of atmospheric temperature fluctuations and experiments on interdecadal climate variability. The emerging scenario reveals





that local variability, e.g. in Western Europe or North America, has increased in the past 40 years. Vincze et al. (2017) investigated the nature of connections between external forcing and climate variability conceptually using a laboratory experiment subject to continuously decreasing 'pole-to-equator' temperature contrast $\Delta T$. Finally, Rodda and Harlander (2020) recently

demonstrated the potential for using laboratory data to study multiple-scale interactions and explain even mesoscale atmospheric processes. Their study reveals that frequency spectra from the differentially heated annulus experiment are comparable to the power spectra from atmospheric field observations.

The paper is structured as follows. In section 2, we briefly describe the experiment, and in section 3, we give insight into typical flow regimes of the annulus experiment. We further study the impact of polar warming on the wave train structure and

zonal phase speed. Section 4 is the core of this paper, where we investigate the temperature distributions and the impact of polar warming on extreme event frequency from experimental data. We then inspect and compare some of these features in NCEP reanalysis data in section 5. Finally, in section 6, we offer our concluding remarks.

## 2 Experimental apparatus and measurements

The experiments presented in this paper have been run with a differentially heated rotating annulus at the BTU Cottbus-

Senftenberg laboratories (see von Larcher and Egbers (2005) for more details about the experimental apparatus). The experimental setup, sketched in Fig. 1, consists of a cylindrical tank divided into three concentric rings, which are filled with de-ionised water. The inner cylinder has a radius $a = 4.5$ cm and is made of anodised aluminium. The water filling the inner cylinder is cooled by an external thermostat connected to the experiment. The tank is made of borosilicate glass, and the outer ring is separated from the middle cavity by a wall placed at radial distance $b = 12$ cm. Heating wires, supplied with constant

power by a control unit, warm up the water in the outer ring. The mid-gap (of width $b - a = 7.5$ cm) is filled up to the height $D = 5$ cm, and the fluid is subjected to a radial temperature difference $\Delta T$ imposed at the boundaries by the two thermally conducting walls. The tank is mounted on a turntable, which rotates counterclockwise around its vertical axis of symmetry at a constant rate $\Omega$. The combined effect of the radial temperature difference produced by the two thermal baths and the rotation of the tank results in the set-in of the baroclinic instability in the middle gap giving rise to baroclinic waves. These waves

developing in the annulus are analogous to the atmospheric Rossby waves, which shape the meandering of the mid-latitude jet stream.

Our experimental data set consists of nine runs that only differ in the cooling temperature set in the inner cylinder, spanning from 9 °C to 18 °C. All the other parameters are kept invariant throughout the different runs. Each experiment is run as described in the following. After setting the temperature in the cold and hot baths, we waited for two hours until the system

reached the thermal equilibrium. The temperature in the outer ring is at a constant temperature that depends on the temperature of the cold bath. During this warm-up time, the tank is at rest. Sensors are placed in the outer warm and inner cold ring to measure the temperature in the two thermal baths, $T_{\mathrm{cold}}$ and $T_{\mathrm{warm}}$, and calculate the radial temperature difference $\Delta T$. Figure 2 and table 1 show the mean temperature measured by these sensors as a function of the temperature set in the cooling basin of the thermostat for each experimental run. Increasing the temperature in the cold bath leads to a higher equilibrium





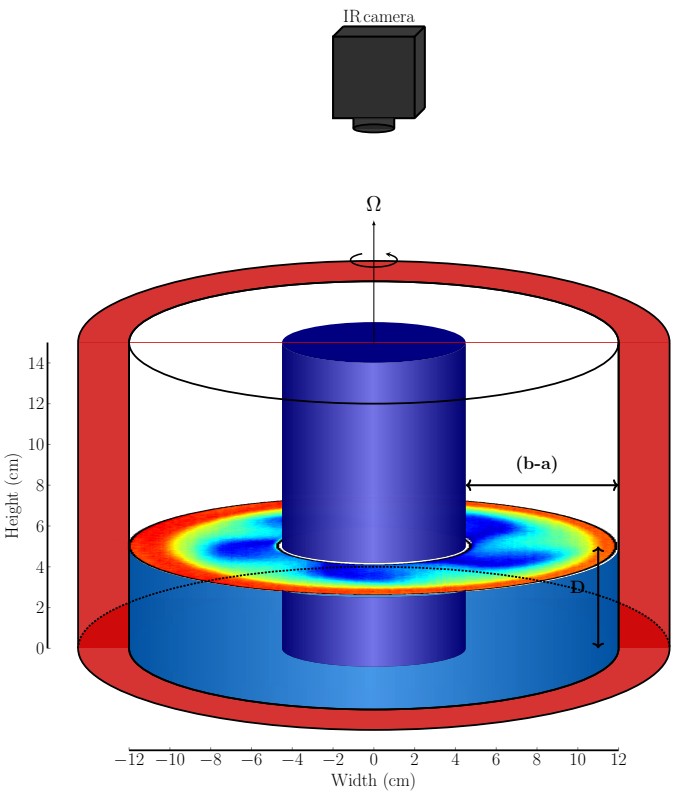

**Figure 1.** Schematic drawing of the experimental apparatus. The water in the inner cylinder (in blue) is cooled using a thermostat. Heating wires heat the water in the outer ring (in red). The water in the inner gap $(b - a)$ is subjected to a radial differential temperature inferred by the insulating walls. The tank rotates counterclockwise at a constant rotation rate $\Omega$. The surface temperature of the fluid in the gap is measured by an infrared camera aligned with the axis of rotation.

temperature and consequently an increase in the temperature in the warm outer bath, which is supplied with the same power at each experimental run. The rise in temperature in both baths is visible in Fig. 2. But the cold ring warming is much higher than the increase in the hot ring. This diverse response to the cooling thermostat change reproduces a warming scenario similar to the Arctic amplification in the experiment. In the rest of the paper, we use the more general term 'polar warming' for this scenario.

After the two-hour warm-up, we start the rotation rate $\Omega$ set to 8 rpm counterclockwise. The system is let run for the spin-up time (approximately two hours), after which it reaches a stable state. Then, the data are collected for seven hours, corresponding to ca. 3000 revolutions of the tank. Given that each revolution corresponds to one terrestrial day, our experiment simulates nine cases with different pole-to-equator temperature differences lasting each for little more than eight years in the lab-Earth analogy.





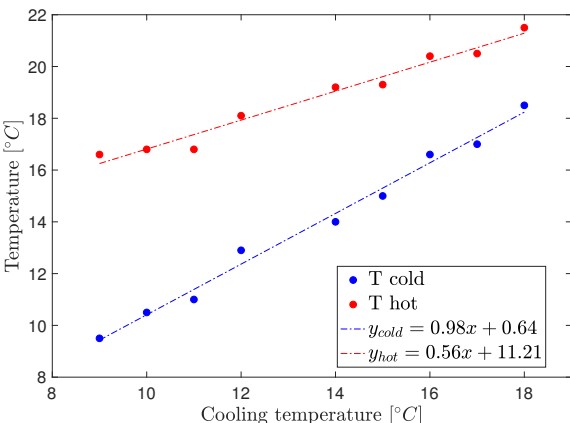

**Figure 2.** Temperature in the cold inner cylinder (blue) and hot outer ring (red) as a function of the cold thermostat target value. In the series of experiments discussed here (see Table 1), the temperature of the cooled water was increased 1.75 times more than the temperature of the heated water. The series of experiments form an Arctic amplification scenario with an increase in mean temperature but a decrease of the radial temperature gradient.

The surface temperature is measured with an Infrared Camera, IR camera in short (Jenoptik camera module IR-TCM 640, with thermal sensitivity of 0.01 K and image resolution $640 \times 480$ pixels). The IR camera is fixed in the laboratory reference system and mounted at the top of the experiment (see Fig. 1). The IR camera outputs are the time series of the entire annulus 2D surface field measured once at each tank rotation, corresponding to a sampling interval $dt = 7.5$ s. The advantage of having

the 2D field is that some characteristics, such as dominant wave patterns, zonal phase speed (i.e. the speed at which the pattern drifts as a whole anticlockwise around the apparatus in the rotating frame), and changes in the structure, can be detected. The spatial and temporal resolution are $400 \times 400 \times 3000$ (pixels and times, $x, y, t$).

## 3   Flow regimes

This section discusses the dependency of the flow regimes on the radial temperature difference $\Delta T$. Understanding how

changes in $\Delta T$ impact the spatio-temporal flow features is a necessary first step for examining the effects on the temperature distribution.

Two nondimensional similarity parameters control the flow behaviour in a differentially heated rotating annulus (Hide and Mason, 1975): the Taylor number

$$Ta = \frac{f^2(b-a)^5}{\nu^2 D}, \tag{1}$$

and the thermal Rossby number

$$Ro_T = \frac{gD\alpha\Delta T}{f^2(b-a)^2}. \tag{2}$$





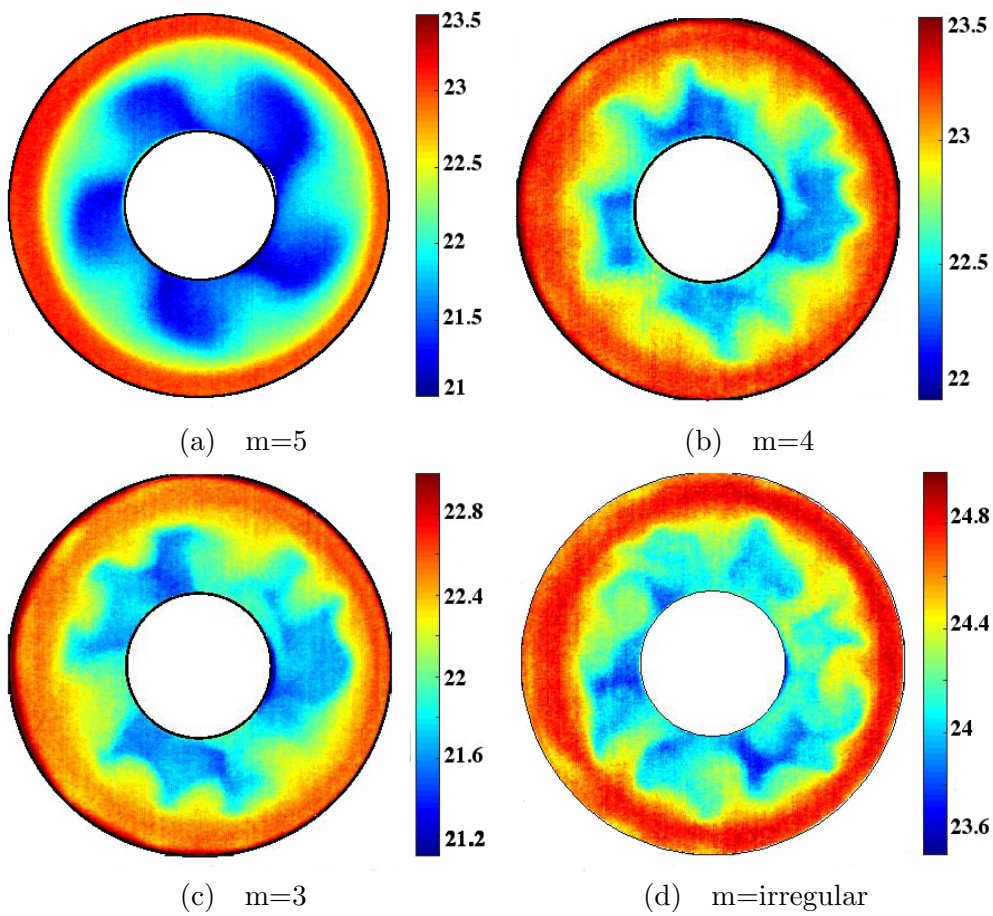

**Figure 3.** Flow regimes at $\Delta T = 7.1K$ (a), $\Delta T = 5.2K$ (b) and (c), and $\Delta T = 3K$ (d). The flow becomes more irregular for decreasing $\Delta T$.

Here, $g$ is gravity, $f = 2\Omega$ is the Coriolis parameter, $\Omega = 8$ rpm the tank rotation, $\alpha = 0.207 \times 10^{-3}$ K$^{-1}$ the volumetric thermal expansion coefficient, $\nu = 1.004 \times 10^{-6}$ m$^2$s$^{-1}$ the kinematic viscosity of water, $D = 0.05$ m the fluid depth, and $b - a = 0.075$ m the gap width. In general, increasing $Ta$ leads to a more turbulent flow. A small $Ro_T$ enhances geostrophy but also forces

2D turbulence. A larger $Ro_T$, on the other hand, leads to more regular wave regimes. Since we kept the tank rotation constant, the Taylor number, $Ta = 1.32 \times 10^8$, is constant as well and $Ro_T \sim \Delta T$.

Table 1 lists the experimental runs and their parameters. $T_C$ is the temperature of the cooling thermostat. $T_{\text{cold}}$ and $T_{\text{warm}}$ are the mean temperatures measured by sensors placed in the inner cold cylinder and warm outer ring, respectively, once a constant temperature difference $\Delta T$ is reached. Due to $T_C$ rising, the mean temperature in the gap $\overline{T}$ (resulting from a thermal

equilibrium between the boundary walls) increases, even though the heating power supply to the outer ring is kept constant.




**Table 1.** Overview of the experimental runs. $T_C$ is the temperature of the cooling thermostat. $T_{cold}$ and $T_{warm}$ are the measured temperatures in the inner and outer ring respectively. $\overline{T} = (T_{warm} + T_{cold})/2$ is the mean temperature. $\Delta T = T_{warm} - T_{cold}$ is the radial temperature difference. $m$ is the azimuthal wavenumber of the baroclinic wave, and $I$ indicates irregular waves. The rotation rate is $\Omega = 8$ rpm for all runs. $Ro_T$ is the thermal Rossby number (2). The Taylor number (1) is $Ta = 1.32 \times 10^8$.

| Name | $T_C$ | $T_{cold}$ | $T_{warm}$ | $\overline{T}$ | $\Delta T$ | $m$ | $Ro_T$ |
|------|-------|-----------|-----------|------|------|-----|--------|
| C18 | 18 | 18.5 | 21.5 | 20 | 3 | I | 0.019 |
| C17 | 17 | 17 | 20.5 | 18.7 | 3.1 | I-3 | 0.020 |
| C16 | 16 | 16.6 | 20.4 | 18.5 | 3.8 | I-3 | 0.024 |
| C15 | 15 | 15 | 19.3 | 17.1 | 4.5 | 4-3 | 0.029 |
| C14 | 14 | 14 | 19.2 | 16.6 | 4.8 | 4-3 | 0.034 |
| C12 | 12 | 12.9 | 18.1 | 15.5 | 5.2 | 4-3 | 0.034 |
| C11 | 11 | 11 | 16.8 | 13.9 | 5.8 | 4-3 | 0.037 |
| C10 | 10 | 10.5 | 16.8 | 13.6 | 6.3 | 4-3 | 0.040 |
| C9 | 9 | 9.5 | 16.6 | 13 | 7.1 | 5-4 | 0.046 |

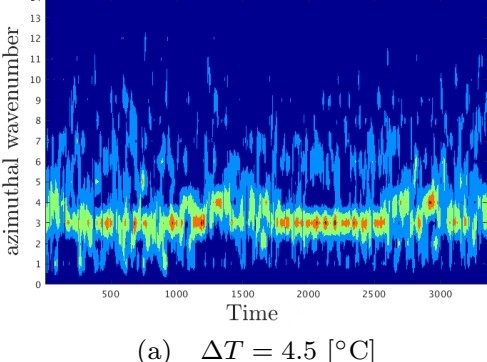
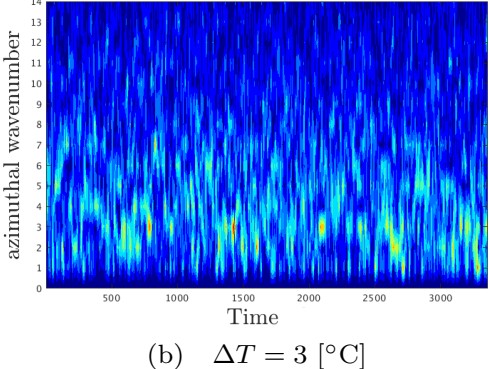

(a)  $\Delta T = 4.5$ [°C]  (b)  $\Delta T = 3$ [°C]

**Figure 4.** Azimuthal wavenumber evolution over time for experiments C15 (a) and C18 (b). The colormap represents the Fourier transform of temperature data sampled at constant radius $r_0 = 0.063$ m. Larger $\Delta T$ are associated to flows with a dominant wavenumber where sporadic transitions occur over time (a). When $\Delta T$ is decreased the flow is irregular as a result of a superposition of wavenumbers (b).

$T_{cold}$ and $T_{warm}$ are plotted in Fig. 2 as a function of $T_C$. It can be noticed that $T_{cold}$ (in blue) increases 1.75 times more than $T_{warm}$ (in red in figure 2). This temperature change is well suited to mimic the polar warming effect observed in the atmosphere experimentally.



### 3.1 Wavenumber transitions

It is enlightening to study the flow regimes in the azimuthal wavenumber space. The column noted with $m$ in table 1 indicates the dominant azimuthal wave number observed during the data acquisition. Roughly speaking, decreasing $\Delta T$ leads to smaller dominant wavenumbers and a more irregular flow. Figure 3 displays surface temperature snapshots taken with the IR camera during different experimental runs and gives examples of flow regimes. Run C9 (Fig. 3 (a)) has a regular flow regime with a regularly shaped baroclinic wave $m = 5$. With smaller $\Delta T$, all the other experiments exhibit more irregular patterns with

wavenumbers $m \leq 4$, which spontaneously transition to other $m$ over time. This transition can be seen in the time evolution of the azimuthal wavenumber, determined by calculating the spatial Fourier transform of the temperature data measured with a sampling rate of $\Delta t = 3.75$ s along a constant radius $r_0 = 0.063$ m. In experiment C15 (Fig. 4 (a)), the flow is for most of the time in a state with $m = 3$ showing sporadic transitions to $m = 4$. In contrast, in experiment C18 (Fig. 4 (b)) the flow is in a noticeably more irregular state, with a broad spectrum of excited wavenumbers. In the latter regime, it is challenging to

identify a dominant azimuthal wavenumber $m$ and, therefore, we refer to this state as irregular (indicated by $I$ in table 1).

### 3.2 Zonal phase speed

The zonal phase speed of the waves can be estimated combining the thermal wind balance $U_T = Ro_T fL/4$ with (2)

$$U_T = \frac{g\alpha D \Delta T}{f(b-a)}. \tag{3}$$

The linear theory by Eady (1949) predicts that the zonal phase speed of an unstable baroclinic wave in a zonal shear flow

$U = U_0 z$ ($U_0$ constant) of depth $D$ is $c = DU_0/2$. It follows that baroclinic Eady waves are non-dispersive, and each wave mode drifts with the same speed. Previous experimental studies (Fein (1973); Vincze et al. (2015)) have observed that the measured zonal phase speed as a function of $\alpha \Delta T/f$ is better fitted by a more general power-law equation

$$c = B(\alpha \Delta T/f)^\zeta, \tag{4}$$

with $B = gD/(b-a)$ a constant depending on the experimental configuration and the exponent $\zeta$ an empirical parameter.

Note that for $\zeta = 1$, we obtain $c = U_T$ given by (3). However, the zonal velocity, $U$, is not simply a linear function of $z$, particularly in experiments with a free surface. Moreover, due to the lateral walls, the mean flow shows also a shear in the radial direction. Finally, rotation and the curvature of the side walls lead to a slight free surface deformation. These effects taken together, introduce a weak dispersion to the Eady waves. Low-frequency vacillations might result from this dispersion even without nonlinear interactions, as discussed in detail by Harlander et al. (2011). Therefore, the zonal phase speed of the

baroclinic wavefront, in our experiment taken as the contourline of the maximum velocity gradient along the radial direction, is challenging to predict, especially in regimes in which the dominant wavenumber is changing over time (see figure 4). In general, the zonal phase speed $c_M$ can be defined as:

$$c_M = \frac{1}{m} \frac{d\Phi}{dt} = \frac{1}{m} \omega = \frac{\lambda}{T_p} \approx U_T, \tag{5}$$




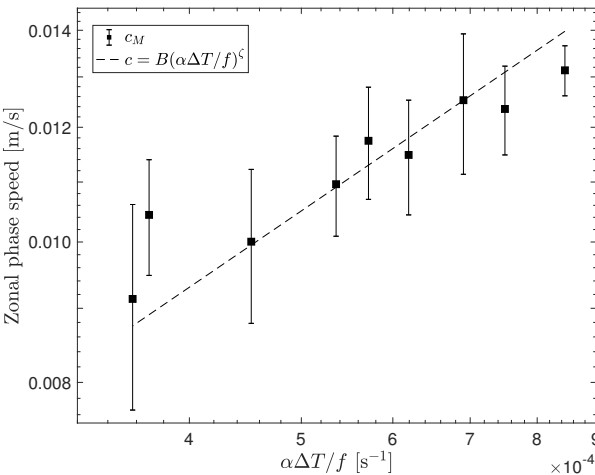

**Figure 5.** Log-log plot of the zonal phase speed $c_M$ of the baroclinic wavefront for decreasing $\Delta T$ calculated from $\phi$-$t$ diagrams at a fixed radius $R_d = 0.087$ m. The dashed line $c$ is calculated from (4) with $\zeta = 0.55$.

where $\Phi = \boldsymbol{k}\boldsymbol{x} - \omega t$ is the phase of a wave, $\omega = d\phi/dt, \lambda = 2\pi/m, T_p = 2\pi/\omega$ are wave frequency, wavelength, and period,
respectively. Therefore, if the measured temperature at a fixed radius $R_d$ is plotted as a function of $\phi$ and time, then the
zonal phase speed can be graphically computed using a Hovmöller diagram. This is done by taking $\Delta\Phi = \lambda/R_d$ and $T_p$ as
vertical and horizontal distances of the wave crests in the plot. For each $\Delta T$, the zonal phase speed is calculated with the
above-explained method for ten-time intervals spanning over the total measurement duration. Then the mean and the standard
deviation are calculated. The mean zonal phase speed $c_M$ with the associated error is plotted in figure 5 as a function of
$\alpha\Delta T/f$. It is evident that decreasing $\Delta T$ slows down the zonal phase speed of the baroclinic waves. The dashed line in figure
5 depicts the general power-law equation (4), where we calculated the coefficient $\zeta = 0.55$ by fitting the data. Figure 5 also
shows that qualitatively the predictions from linear wave theory carry over to large-amplitude nonlinear waves.

Differences between the theory and the real flow notwithstanding, the zonal phase speed $c$ predicted by (5) is close to the
experimentally measured $c_M$, particularly for the regular flow regimes with a zonal wavenumber $m = 4$ wave (the intermediate
values of $\Delta T$ in runs C11-C15, see table 1). Outliers for smaller $\Delta T$ are consistent with the occurrence of dispersion. Compared
with Eady waves, dispersive Rossby waves are always slower than the mean background flow. This holds in particular for long
waves. Thus, from the zonal phase speed results, we can conclude that the effect of decreasing the radial temperature difference
is to slow down the baroclinic wavefront. Hence, the propagation speed of extreme events embedded into the wave trains, e.g.
in the form of exceptional large meanders, will be slowed down, and the events in real atmospheric flows might unfold their
local destructive potential over a more extended period.





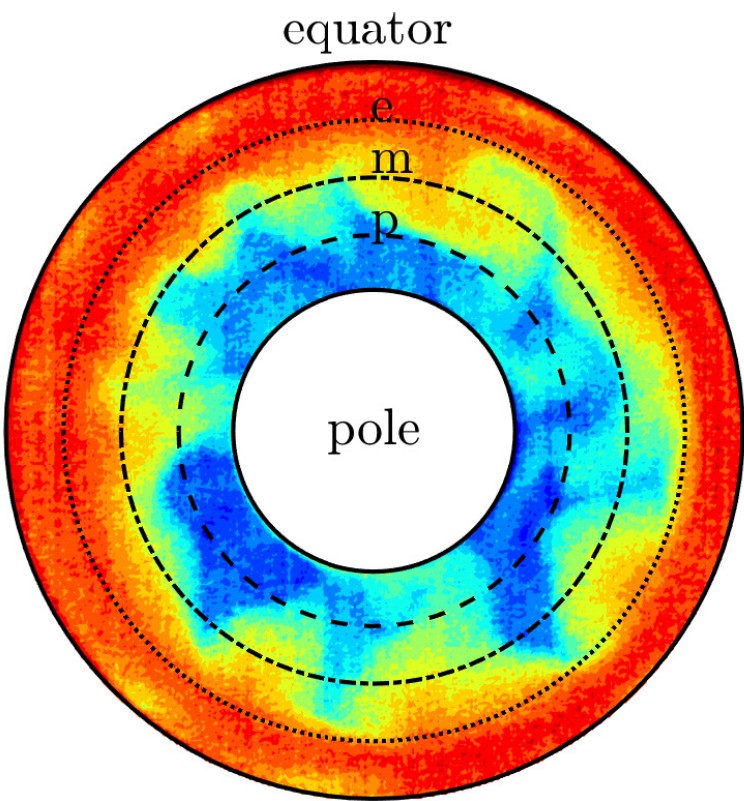

**Figure 6.** Three regions of the domain and location of the baroclinic wave: polar region (between the inner ring and the circle marked with 'p'), mid-latitudes (between 'p' and 'm'), and equatorial regions (between 'e' and the outer ring). The colours show the temperature measured for the experiment at $\Delta T = 4.5$.

## 4   Temperature distributions and variability

This section investigates how the polar amplification affects temperature distributions, particularly focusing on the variability.

Figure 7 compares the probability density distributions of the temperature anomalies for all nine experimental runs. Temperature anomalies are calculated by subtracting the mean temperature from the measurements at each location 'p', 'm', and
'e'. Distributions at all three locations present a visible deviation from Gaussianity, which is more pronounced for larger $\Delta T$. Noticeably, the polar and middle regions are characterised by broader temperature distributions, whilst the distribution in the outer region is much narrower. This difference tends to be less marked for smaller $\Delta T$. For the polar and middle regions (left and middle plots), temperature variability consistently decreases for decreasing $\Delta T$ (red lines), with distributions that become narrower and more symmetric. In the equatorial regions, the tendency is the exact opposite: the distributions broaden for de-





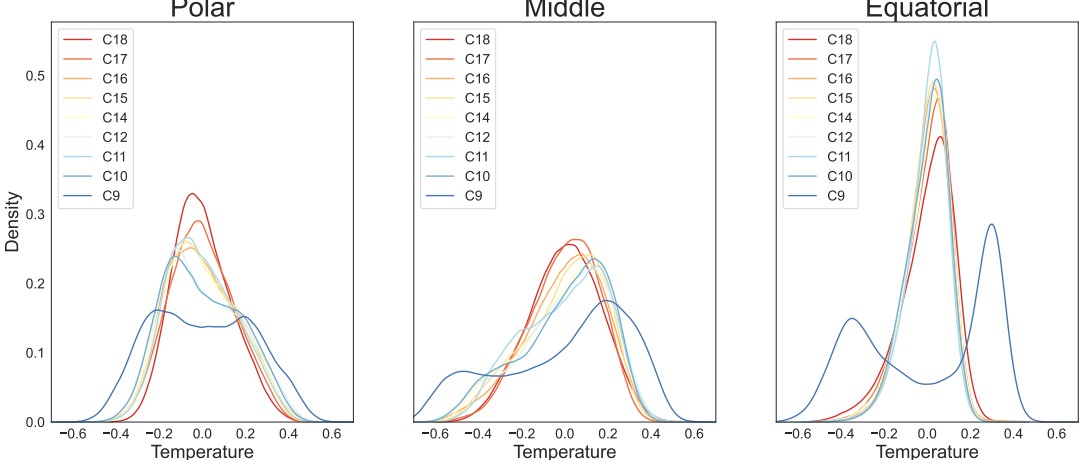

**Figure 7.** PDFs of temperature anomalies for the 9 experimental runs at different $\Delta T$ calculated at fixed latitudes 'p' (left), 'm' (middle), and 'e' (right), see figure 6 for the regions in the annulus.

creasing $\Delta T$. The run "C9", at the highest $\Delta T$, stands out for its double-peaked distributions at all latitudes, which are also much broader than all other runs.

The changes in temperature anomalies variability are quantified in Fig. 8, where standard deviation (left), skewness (middle), and excess of kurtosis (right) are plotted for the distributions at the three latitudes shown in Fig. 7.

The standard deviation decreases with decreasing $\Delta T$ in the polar and mid-latitude regions (blue and black lines), whilst it increases in the proximity of the equator (red line). To establish the statistical significance of the trends, we calculated the $p$-value with the $t$-test of the fit linear regression model `fitlm`, considering the 95% significance level. Hence, we consider trends with $p < .05$ as statistically significant, meaning that there is a probability higher than 95% that the corresponding coefficient is different from zero.

The negative trends of the standard deviation are statistically significant ($p = .01$ for the polar region and $p = .003$ for the middle region), but the positive trend in the equatorial region is nonsignificant ($p = .11$). Note that the data point corresponding to the highest $\Delta T$ has been ignored to calculate the trends in Fig. 8 since it lies far off all the other data points. These points correspond to the double-peaked distributions in Fig. 7.

The excess kurtosis in Fig. 8 (right) shows an evolution towards more Gaussian distributions for decreasing $\Delta T$, confirming what can be seen in Fig. 7. The polar and middle regions have a highly significant positive trend ($p < .001$ in both cases). In the equatorial region, the excess kurtosis has a nonsignificant negative trend ($p = .5$).

The skewness (Fig. 8 middle) is relatively robust under changes in $\Delta T$: the polar data possess a right (positive) and the equatorial data a left (negative) skew without any significant trend. This opposite sign for the skewness is also observed in Garfinkel and Harnik (2017) in the case of measurements of near-surface tropospheric temperatures. Moreover, the skewness





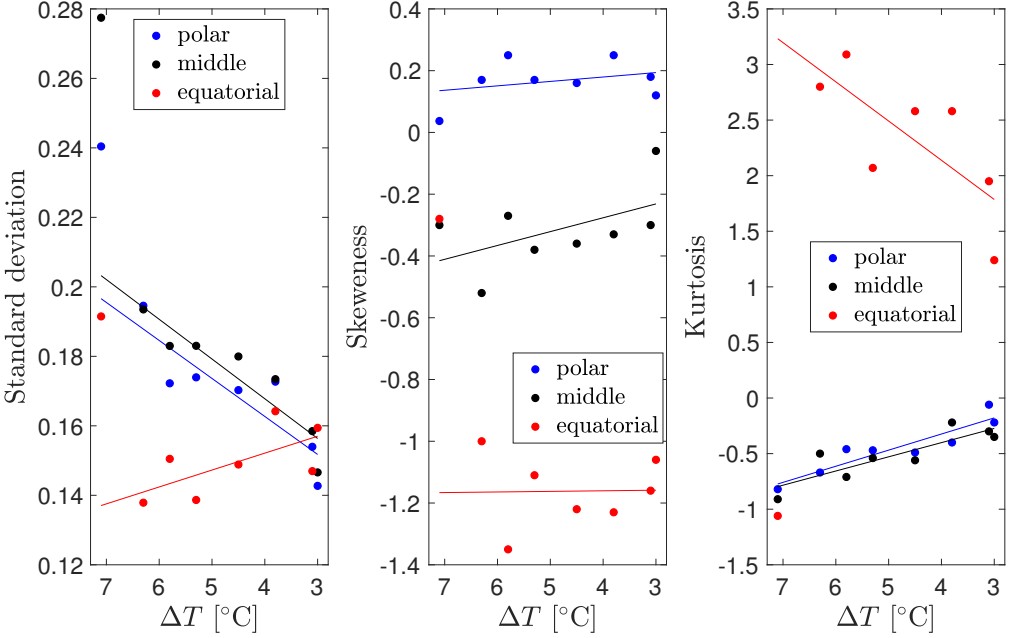

**Figure 8.** Standard deviation, skeweness and excess kurtosis trends for decreasing $\Delta T$ for the inner most ring (blue dots), the middle ring (black dots), and the outer ring (red dots).

behaviour reminds us of the results by Belmonte and Libchaber (1996), who found for turbulent Rayleigh-Bénard convection

that the temperature distribution skewness has a positive value at the cold (top) boundary and becomes more and more negative close to the warm (bottom) boundary.

The skewness (Fig. 8 middle) does not show a statistically significant trend for any of the regions investigated. Linz et al. (2018) studied the effects of decreasing $\Delta T$ on temperature distributions in an idealised advection-diffusion model and concluded that whilst a smaller $\Delta T$ reduces the variance, it does not have any direct effect on the skewness and kurtosis. Changes

in the kurtosis should be attributed to a response to changes in the flow field instead. Therefore, the lack of a clear trend in skewness in our data is in agreement with the results by Linz et al. (2018).

Our experimental data clearly show a relation between the decrease in meridional temperature difference and temperature variability. These results are consistent with what was found by Dai and Deng (2021) in model simulations and reanalysis data. Their analysis indicates that Arctic amplification decreases the temperature variability over the northern mid-high latitudes.

What does the reduction in temperature variability mean in terms of extreme events? Narrower and more symmetric probability distributions imply that cold and warm extreme events become weaker with decreasing $\Delta T$. It follows that extreme events decrease in intensity in polar and middle regions whilst near the equators, they become stronger. However, this does not help predict extreme events' duration or frequency. In the next section, we analyse the impact of changes in $\Delta T$ on the frequency of extreme events.



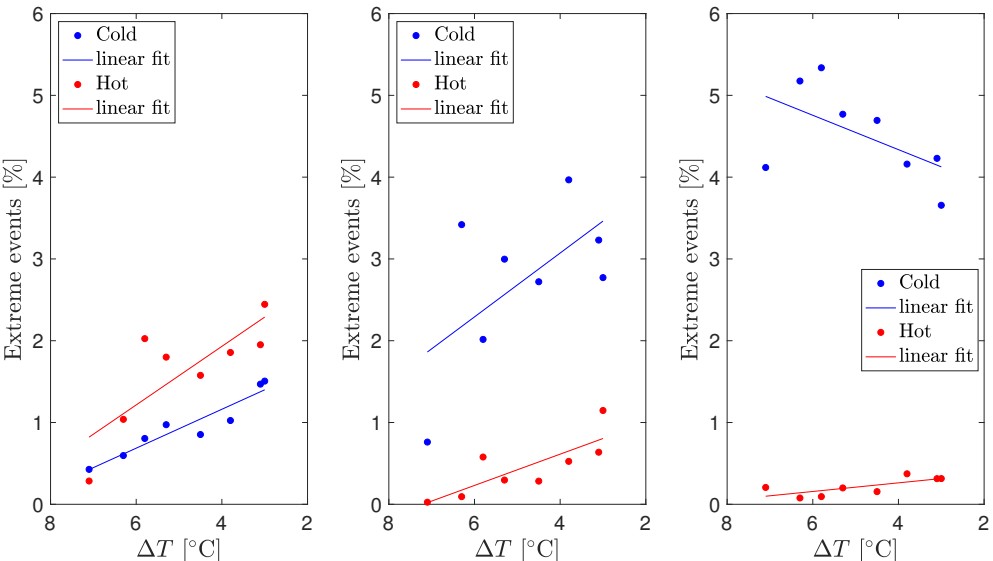

**Figure 9.** Trend of the cold (blue) and hot (red) extreme events with decreasing equator-pole temperature difference. The trends are considered at three latitudes in the plot: polar latitude (left), mid-latitude (middle), and equatorial latitude (right). The three latitudes are indicated in figure 10 by the labels 'p', 'm', and 'e', respectively.

## 4.1 Extreme event frequency

Extreme events are defined using a variety of metrics such as temperature thresholds and indices. The thresholds can be defined in different ways, the most common distinction being between relative thresholds (for example, defined by using specific percentiles of distribution or more straightforward measures like standard deviation) or absolute thresholds (for example, days with temperatures exceeding 35°) (Seneviratne et al., 2021). Therefore, the choice of the definition used to calculate extreme events can affect the meaning of extremes and possibly the results.

We have discussed in section 4 that the reduction in meridional temperature difference has an impact on the temperature variability and results in milder extreme temperatures from the poles to the mid-latitudes, whilst at low latitudes, the extremes might become stronger. To study whether this variability reduction impacts the frequency of extreme events, we define extreme temperature events based on a relative threshold. We chose such threshold as the standard deviation ($\sigma$) calculated for each experimental run, corresponding to a fixed $\Delta T$, on ensembles of data measured at fixed latitudes. We then call extreme cold/hot events all temperatures such that $|T - T_{\mathrm{mean}}| > 2\sigma$.

Note that the $\sigma$−threshold is latitude-dependent. The extreme event frequency is defined as the number of times the temperature crosses the set threshold normalised by the total length of the data set (which is, in any case, the same for all the data sets considered). The event duration is neglected, i.e. only the number of measurements ("days") where the temperature exceeds the threshold is counted, without distinguishing whether such days are consecutive or isolated.





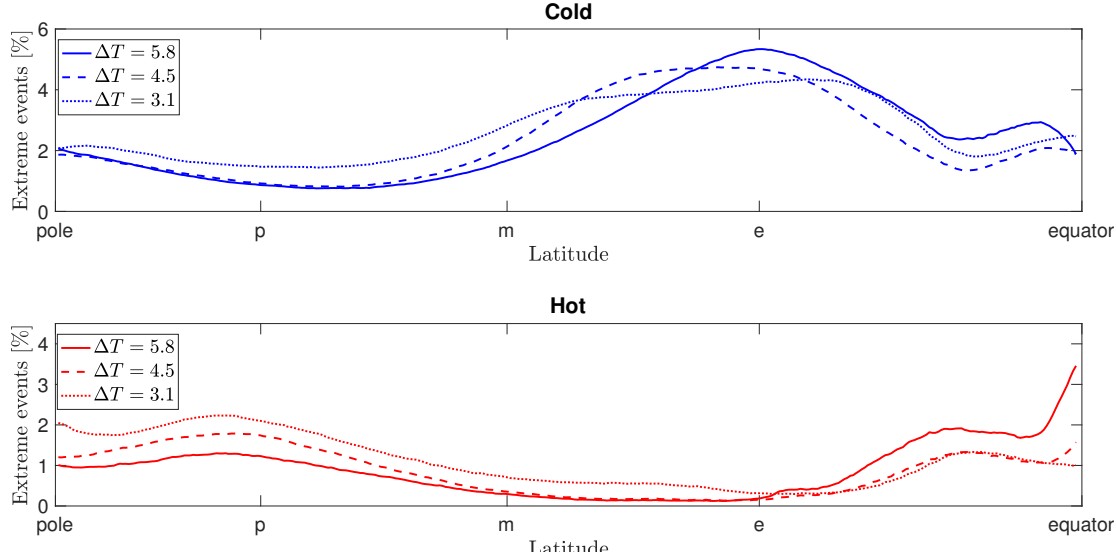

**Figure 10.** Number of extreme event (in percentage) as a function of latitude in the experiment. The upper plot depicts the cold events, whilst the bottom plot depicts the hot events. The lines are: solid for the experiment at $\Delta T = 5.2K$, dashed at $\Delta T = 3.8K$, dash-dot line at $\Delta T = 3K$. The labels 'p', 'm', and 'e' indicate the latitudes at which the trends shown in figure 9 are taken (see figure 6 to visualise their position).

The number of extreme events as a function of $\Delta T$ is plotted in Fig. 9 at the three locations 'p' (in the left plot), 'm' (in the middle plot), and 'e' (in the right plot). The cold events are plotted in blue and the hot events in red. By comparing the three plots, one can immediately notice that the number of extreme cold and hot events is similar in the region near the pole (left figure). Still, the extreme cold events become more and more frequent moving towards the equator (middle and right plots).
The second noticeable feature is that the hot extreme events show a statistically significant trend for all three regions ($p = .016$ in the left plot, $p = .016$ in the right plot, $p = .036$ in the right plot in figure 9. This gives evidence for a hot extreme events increment for decreasing $\Delta T$. For the cold extremes, the left plot reveals a statistically significant positive trend ($p < .001$), the middle plot a positive trend as well, but it is statistically nonsignificant ($p = .055$), whilst the right plot shows a statistically nonsignificant negative trend ($p = .08$).
This difference in the cold trends can be partially explained by the fact that the baroclinic wave dynamics do not govern the equatorial dynamics, and we have already suggested that the extreme event distribution along the inner regions is tightly linked to the baroclinic waves. The decrease in cold events is consistent with a possible change in the equatorial extension of the baroclinic wave.

For a more complete understanding, we also study how extreme events are distributed as a function of latitude. For this
analysis, temperatures at fixed latitudes are collected into sets, where each set is constituted by $1.2 \times 10^6$ temperature measure-





ments for different times and longitudes. The standard deviation $\sigma$ is calculated for each set, and, successively, the extreme event frequency is calculated as a function of the latitude.

The distributions are plotted in Fig. 10 for three experiments, namely at $\Delta T = 5.2K$ (solid line) $\Delta T = 3.8K$ (dashed line) and $\Delta T = 3K$ (dashed-dotted line). For easier visualisation, these three experiments have been chosen to represent the entire

data set. The excluded data show similar results. The upper plot presents the extreme cold events, whilst in the lower plot we see the extreme hot events. Three latitudes labelled as 'p', 'm', and 'e', are indicated on the $x-$axis for better readability.

For both plots, we can notice that the three curves have similar shapes from the polar latitudes up almost to the point 'e', with more frequent extreme events for decreasing $\Delta T$. Yet, the cold and hot extremes exhibit opposite trends: the cold event frequency increases from the pole towards the equator with a maximum just before 'e' and then decreases, whilst the extreme

hot events behave the opposite. To some extent, the existence of a local maximum for warm and minimum for cold events at latitude 'e' can be understood by looking at figure 6, where the latitude 'e' (dotted line) marks the limit of the extension of the baroclinic wave cold tongues, covering approximately three quarters of the gap width. The external ring (between 'e' and the 'equator' in the figure) is unreached by the cold tongues; therefore, we can expect differences concerning extreme event frequency in this part of the annulus where we find no baroclinic wave activity. The location of the maximum/minimum

in cold/warm event frequency coincides with the utmost latitude to which the baroclinic waves extend, and this is a clear indication that the baroclinic wave activity shapes the frequency distribution.

In summary, we suggest that the large-scale baroclinic wave dynamics govern the extreme event spatial frequency distribution in the laboratory experiment.

## 5 High-variability events in NCEP data

After analysing the lab data for changes in variability and the distributions of high-variability events as a function of the North-South temperature gradient, it is instructive to see what atmospheric data show concerning changes in variability and variability extremes. We use the National Centers for Environmental Prediction (NCEP) reanalysis data (Kalnay and collaborators, 1996). In contrast to operational counterparts, the reanalysis data do not suffer from inhomogeneities introduced by changes in the data assimilation system. In this respect, they are a good supplement to data based on individual instrumental records or climate-

model simulations (Uppala et al., 2008). Moreover, reanalysis data cover historical data as well. We use two temperature data sets from the collection "NOAA-CIRES 20th Century Reanalysis, version 2, Daily Averages", covering the period from 1871 to 2012. The first set is the daily ensemble mean pressure level data (1000hPa to 10hPa), from which we extracted just the 500hPa level. The second set is the daily ensemble mean tropopause data. We start by considering the gradient and the standard deviation of the 500hPa temperature from 1871 to 2012.

The upper panel of Fig. 11 shows the trend in the North-South temperature difference ($\Delta T$) taken from the NCEP 500hPa data for the Northern Hemisphere (blue line) and the Southern Hemisphere (red line). These temperature gradients are evaluated from the zonal mean temperature at $88°N$ ($88°S$) and the equatorial zonal mean temperature. We see that there is no large change for the 500hPa data in the Northern Hemisphere, except for a decline starting from the year 2000. In contrast,





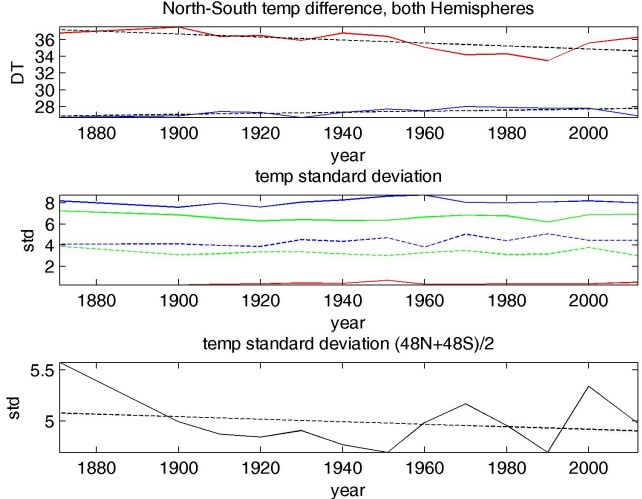

**Figure 11.** NCEP reanalysis data of 500hPa temperature. Upper figure: $\Delta T$ trend from 1871 to 2012, northern Hemisphere (blue), southern Hemisphere (red). The linear regression (dashed line) shows a trend towards smaller North-South temperature differences in the southern Hemisphere ($T(2012) - T(1871) = -2.51K$) but a slightly larger difference for the northern Hemisphere ($T(2012) - T(1871) = 0.88K$). Center figure: zonal mean temperature standard deviations for $88°N$, $88°S$ (blue solid, blue dashed), $48°N$, $48°S$ (green solid, green dashed), and equatorial (red). Bottom figure: mid-latitude zonal mean temperature standard deviation of $((T(48°N) + T(48°S))/2)$ (solid line) and linear regression (dashed line). There is a weak trend towards decreasing standard deviation($\sigma_T(2012) - \sigma_T(1871) = -0.18K$).

for the southern Hemisphere, there is a long-term negative trend. Despite some variability, the linear regressions (dashed lines) show a statistically significant trend towards more considerable North-South temperature differences for the northern ($T(2012) - T(1871) = 0.88K$, $p = .042$) and smaller North-South temperature differences for the Southern Hemisphere ($T(2012) - T(1871) = -2.51K$, $p = .021$). It is surprising at first glance that we do not see more clearly the warming of the Arctic (Arctic amplification) in the Northern Hemispheric data. However, the warming is pronounced particularly in surface data. For the lower stratosphere, the Arctic region is even cooled due to climate change (Stendel et al., 2021) and, as shown in the upper panel of Fig. 11, according to the NCEP data even for the 500hPa level, the warming is not obvious. In fact, this discrepancy between the change of the North-South temperature gradient for low and high levels of the troposphere drives the debate whether climate change leads to more or less wavier jet streams (Stendel et al., 2021).

The central figure displays the time evolution of the temporal standard deviation $\sigma_T$ at different latitudes (blue solid, $88°N$; blue dashed $88°S$; green solid, $48°N$; green dashed, $48°S$; red, equator). The Northern Hemisphere exhibits a more significant variability, which might be related to a stronger land-sea contrast. For the mid-latitude time series, a small temporal decline can be seen (green lines), and the course of the curves looks very similar for both Hemispheres. In the bottom figure, we highlight this trend by plotting just the mid-latitude zonal mean, the standard deviation of $(T(48°N) + T(48°S))/2$ (solid line) and adding the linear regression (dashed line). Obviously, there is a weak but statistically not significant trend toward decreasing standard deviation ($\sigma_T(2012) - \sigma_T(1871) = -0.18K$, $p = .49$). Though not significant, this trend is consistent with





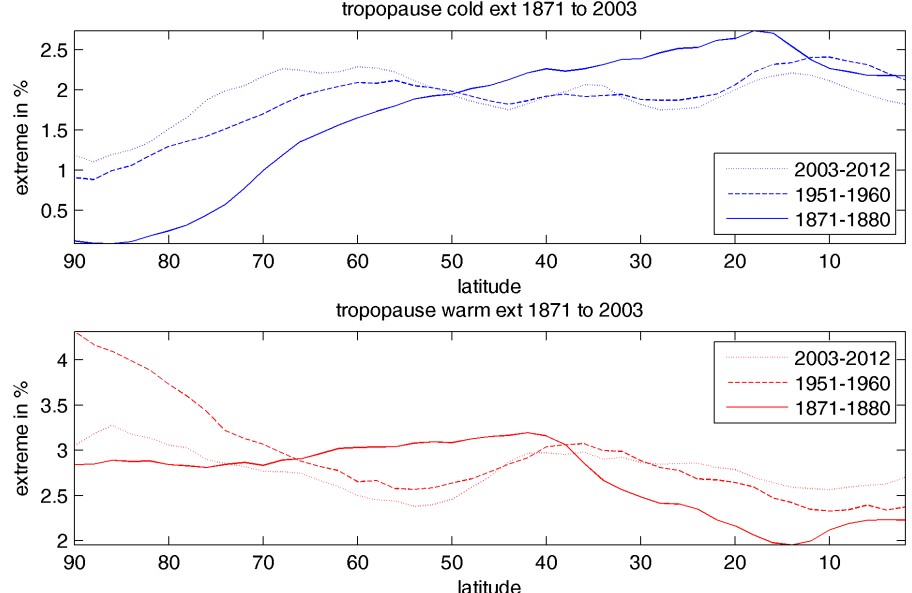

**Figure 12.** NCEP reanalysis data of tropopause temperature. Tropopause data have been chosen since the frequency spectra of the baroclinic wave experiment look most similar to tropopause spectra (Rodda and Harlander, 2020). A number of extreme events in % for different time slices and latitudes. The upper and bottom plots depict extreme cold and hot events, respectively. The lines are: solid for 1871-1880, dashed for 1951-1960, dotted for 2003-2012. The seasonal cycle has been removed from the data. The extreme values have been taken from the Pacific sector (weak land-sea contrast), and the frequencies from the southern and northern Hemisphere have been averaged for any latitudinal circle.

other model data (Rind et al., 1989). These observations are also in qualitative agreement with the results from the laboratory experiment.

To inspect the frequency of high-variability (extreme) events, we focused on NCEP reanalysis data of the tropopause temperature. These data are less prone to the effects of land-sea contrasts and might be closer to the lab experiment. Rodda and Harlander (2020) have shown that spectra from tropopause data are comparable to the frequency spectra of the baroclinic wave experiment, and we suggest a similar connection concerning extreme values. We highlighted three ten year periods: from 1871 to 1880, from 1951 to 1960, and from 2003 to 2012. The seasonal cycle has been removed from the data. The frequency of extreme values, defined as values larger or smaller than twice the standard deviation, has been calculated for all longitude circles with an increment of $20°$. Subsequently, for enhancing the robustness of the analysis, we took the mean of the Northern *and* Southern Hemisphere frequencies and finally we zonally averaged the frequency data. This gives the mean frequency of extreme values as a function of latitude ranging from $90° - 0°$.

We see from Fig. 12 that the number of extremes, broken down to values above (warm, red curves) or below (cold, blue curves) the $2\sigma$ standard deviation threshold, ranges between 1 and 4%. For the cold case, the frequency of extreme events is



larger at low latitudes, i.e. 1.7% to 2.5% in the latitude band from $30°$ to $10°$ and 0.25% to 2% in the latitude band from $80°$ to $60°$. We can see that the period 2003-2012, with the smallest North-South temperature gradient, shows most extremes for polar latitudes (dotted blue line, $90°$ to $60°$) but least for tropical latitudes (dotted blue line, $20°$ to $0°$). Note that the experimental data lack of a tropical dynamics (f-plane with a wall as a southern boundary). Hence, one must be careful when comparing low-latitude NCEP data results with experimental data close to the heated outer sidewall. For the warm extremes, we see the highest frequency for the period 1951-1960 (dashed red line) near the polar region (more than 4%). In this polar region, the most recent period shows more extreme warm events (dotted red line) than the historical data (solid red line) but about 1% less than the period 2003-2012 (dashed red line). Obviously, for the frequency of extreme events, the strongest North-South gradient can be found for the decade 1951-1960 (dashed red line).

Comparing the experimental data (Fig. 10) with the NCEP data (Fig. 12) we find a notable qualitative similarity of the curves from 'p' to 'e' and from about $90°$ to $20°$, respectively. Cold/warm extreme events occur more at low/high latitudes, in agreement with what was observed by Garfinkel and Harnik (2017). Furthermore, the cold events in NCEP and experimental data with large $\Delta T$ are most numerous at low latitudes. The frequency decreases for lessening $\Delta T$. In contrast, cold events are most probable for smaller $\Delta T$ for near polar and equatorial latitudes. It should be noted that the NCEP data (Fig. 12) do not show such distinct extreme event peaks around the near equator region but have a monotonic increase/decrease for cold/hot events instead. This difference can be expected since the appearance of a local peak in the experiment might be due to the mentioned missing tropical dynamics, as previously explained. Note further that the largest $\Delta T$ experiment gives a very regular baroclinic wave (see figure 3 a), somewhat unrealistic with respect to atmospheric flows. Caution is, therefore, required when comparing the $\Delta T = 5.2$ experiment with atmospheric temperature data.

## 6 Conclusions

We have presented a series of experiments with a differentially heated rotating annulus to model a global warming scenario. Our study aims to reproduce the Arctic warming and study the possible effects on other atmospheric phenomena. For this simple experimental environment, the impact of a reduced pole-to-equator temperature difference on the mid-latitude large-scale dynamics and consequences for the likelihood and distribution of extreme events could be isolated from various processes that, in a more complex way, might play a role also for the Earth's atmosphere.

We found that the jet stream becomes more irregular due to warming the pole, making it challenging to identify a clear dominant azimuthal wavenumber. Moreover, the eastward propagating speed of the meandering baroclinic jet decreases, which, for the experiment, is a consequence of a slow down of the westerly mean flow for reduced $\Delta T$. The decreasing meridional temperature difference also leads to a reduction of the temperature variability in regions of the experiment where the baroclinic waves drive the dynamics. These regions, corresponding to polar and mid-latitudes are characterised by temperature distributions that become narrower and more symmetric. Towards the outer ring of the annulus, corresponding to equatorial regions, the dynamics is less affected by the baroclinic, and consequently the variability shows a slight increase. Our experimental findings are in agreement with the recent analysis of coupled model simulations and ERA5 reanalysis by Dai and Deng (2021).

The consequence of a reduced temperature variability is that extreme events tend to become milder. However our experiment also reveals that extreme cold/warm events tend to become more frequent. Furthermore, these events are larger in number at lower/higher latitudes independently of the time period and temperature differences considered in agreement with NCEP data and with measurements of near-surface tropospheric temperature reanalysis data (Garfinkel and Harnik, 2017).

We think the results of the study underpin the usefulness of the laboratory approach to understanding specific processes of climate change, in particular with a view to temperature variability and extreme events. However, we have only taken a first step and more sophisticated aspects like the use of extreme value theory, long term memory effects, heavy tails in the amplitude of fluctuations, power-laws, spatial correlations and teleconnections etc., have been neglected in the work presented here. Moreover, a more recent and bigger rotating tank has proven to be closer to the atmospheric case than the smaller system

used here (Rodda et al., 2020). Hence, further experiments using this bigger differentially heated rotating tank and a deeper statistical analysis are planned for the future to add more experimental data to observations and climate simulations.

*Data availability.* The NCEP reanalysis data are from https://psl.noaa.gov/data/gridded/data.20thC_ReanV2c.pressure.html. We used two temperature data sets from the collection "NOAA-CIRES 20th Century Reanalysis, version2, Daily Averages", covering the period from 1871 to 2012. The first set is the daily ensemble mean pressure for the 500hPa level. The second one is the daily ensemble mean tropopause

data set.

*Author contributions.* All authors have contributed in conceiving and designing the experiments. C.R. has run the experiments and analysed the experimental data. U.H. has analysed the NCEP data. C.R. and U.H. have drafted the paper. All authors have reviewed the manuscript and given final approval for publication.

*Competing interests.* The authors declare that they have no competing interests.

*Acknowledgements.* This work was supported by the *Spontaneous Imbalance Project* (HA 2932/8-1 and HA 2932/8-2), that is part of the research group *Multiscale Dynamics of Gravity Waves* funded by DFG (FOR1898). The work of M.V. was supported by the National Research, Development and Innovation Office (NKFIH) of Hungary under grant FK125024.



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
