# Peer review of "Jet stream variability in a polar warming scenario - a laboratory perspective"

_EGUsphere, 2022_

## Author Comment (AC1)

(a)                                                          (b)

---

## Author Response (AR1)

**COMMENTS**

**To:** Referee 1

**From:** Costanza Rodda & Co-authors

**Subject:** Responses to the review

We thank the referee for the valuable comments and efforts towards improving our manuscript. In the following, we comment (italic font) on the referee's suggestions and questions point by point. Modifications are marked in red in the revised manuscript.
* * *
I found Rodda et al. 2022 to be relevant, interesting and nicely presented work. Addressing my comments below will mainly improve its presentation and clarity.

1. **Referee** Line 50. How does an atmospheric Rossby wave (ARW) differ from a thermal Rossby wave? Further, how are mid-gap baroclinic instabilities analogous to ARW's when this experimental system contains no beta-effect?

   **Reply** *The thermal Rossby wave is discussed for experiments with very fast rotation. In that case, centrifugal force becomes strong compared to gravity and a strong radial buoyancy develops. Such experiments are outside the regime of atmospheric baroclinic instability but are relevant for the formation of convective columns in stars or fluid planet cores (see e.g. Busse and Or 1986, Convection in a rotating cylindrical annulus: thermal Rossby waves, JFM). In contrast, the differentially heated rotating annulus is a simplistic model that captures the essential dynamical processes driving the circulation of the terrestrial atmosphere at mid-latitudes, particularly the large-scale waves and the meandering of the jet stream. Such a laboratory model has been extensively used to understand the mechanism generating large-scale atmospheric waves (baroclinic waves): the so-called sloping convection (see Lorentz 1967, Hide and Mason 1974). The jet stream is driven by baroclinic instability which does not depend on $\beta$-plane effects. The most simple model for linear baroclinic instability is the Eady model where $\beta$-effects are excluded. Similarly to theoretical models aiming to capture the most essential aspects of the dynamics, the differentially heated rotating annulus experiment does not consider, in its most used configuration, other processes that bring secondary effects into the dynamics, among which is the $\beta$-effect. Another type of laboratory experiment, consisting of a rotating cylinder with sloping boundaries, can be used to investigate $\beta$-effects and generate topographic barotropic Rossby waves. The addition of such sloping bottom in the baroclinic set-up has been studied, even if the analogy with the planetary $\beta$-effect is no longer exact. In general, the sloping bottom modified experiment is reported to excite more complex dynamics with the formation of multiple jets in some cases and direct and nonlocal exchange of energy between the eddies and the zonal modes (see Wordsworth et al. 2008). We plan to do further experiments including beta-plane effects in the future.*
   *We have re-written the paragraph to better explain the analogy between the experiment and the atmospheric waves.*

2. **Referee** Line 80. What is the rotational Froude number for these experiments? Is it far less than unity?

   **Reply** *The centrifugal Froude number, defined as $Fr = \Omega^2(b-a)/g$, where $(b-a)$ is the gap width, is in the order of $5 \times 10^{-3}$ and hence much smaller than 1.*

3. **Referee** Line 85. It says that the temperature in the outer gap is held fixed, but elsewhere it says the heating power is held fixed. So what is actually fixed? A time series of the bath temperatures showing thermal equilibration and then the final experimental phase would make all this clearer to the reader.

   **Reply** *The heating power is held fixed. The system is then let to reach thermal equilibrium with the outer and inner baths setting to a constant temperature that is maintained throughout the entire duration of the experiment. We have added an appendix with details about this and figures showing the time series of the temperature bath, as suggested.*

4. **Referee** Line 119. Why does increasing $Ta$ lead to a more turbulent flow? How do you define turbulent? Is it the value of the Reynolds number or is it the variability / irregularity of the flow field? Typically, higher $Ta$ gives more organized rotating flows. Here, I think it makes the Rossby radius smaller, eventually allowing for multiple structures to fit within the fluid gap. Once this can occur more complex solutions can develop. Is that correct?

   **Reply** *Rotation makes the flow more organised since it becomes more 2D. However, 2D turbulence can develop in such flows and this kind of turbulence is relevant in our experiment. Assuming an aspect ratio of about 1, $Ta = (1/E_k)^2$, where $E_k$ is the Ekman number. For small $E_k$, viscous damping is small compared to advection and waves and vortices can propagate larger distances. For geostrophic turbulence, where the Rossby number has to be small, the Ekman number needs to be small too and hence $Ta \gg 1$. Increasing $Ta$ leads the flow to regimes where the waves undergo amplitude and/or structural vacillation, i.e. they show more irregular features with respect to time and space until there is a regime transition to geostrophic turbulence (as shown by other studies such as Hide and Mason 1975). We used "turbulent", similarly to what is often done in the literature, to indicate the irregular regime, which is analogous to geostrophic turbulence. It is correct that the Rossby deformation radius is reduced and indeed by only increasing rotation, experimentally one observes an increase in the azimuthal wave number in the regular baroclinic wave regime. However, our study focuses on high $Ta$ and low $Ro_T$, which corresponds to a regime where the waves are in an irregular state (or very close to the transition). In this case, the behaviour of the waves is much more complex as nonlinear interactions between the dominant wavenumber and the sidebands occur. We have expanded the section explaining the regimes in more detail.*

5. **Referee** Table 1 appears at odds with my arguments above. The irregular m cases correspond to lower m values. Why would lower m be more complex? Also, why is lower $Ro_T$ corresponding to fewer structures? I would expect from a Rossby radius argument that this would be just the opposite of what is reported here. Again, please explain this in more depth.

   **Reply** *The main consequence of decreasing $Ro_T$ is that the flow becomes more complex due to several wave interactions, as discussed above. The spectra become wider, and it is more difficult to associate a dominant wavenumber to characterise the flow. We agree that Table 1 is potentially misleading the reader and have changed it to show a range of the most energetic wavenumbers for the lower $Ro_T$.*

6. **Referee** Line 158. It is not clear how the $c_M$ equation is used to generate the data in Figure 5. It is a bit opaque as to how the thermography data is processed to measure $d\Phi/dt$. How is this done, operationally? —Further, it is not clear why $c_M \approx U_T$.

**Reply** *We have rewritten the text in the hope that it is now clear to the reader how $c_M$ is calculated from the data. To calculate $c_M$ from the temperature data, we have extracted the temperature along a fixed radius $R_d$ and plotted them as a function of time. In this way, we obtain a Hovmöller diagram with the azimuthal coordinate $\phi$ on the $y-$axis and time on the $x-$axis from which we can calculate the zonal phase speed graphically. The wavefronts appear in this plot as tilted lines of maximum temperature, and their slope is the zonal phase speed at which the baroclinic wave is travelling. We obtained the drift speeds by measuring the slope of ten wavefronts and then calculating the mean value and the associated standard deviation. This procedure is repeated for each experimental run associated with a different $\Delta T$.*

*If we use the linear theory proposed by Eady, the waves can be assumed to be travelling with the same speed of the background flow and therefore $c_M \approx U_T$.*

7. **Referee** Line 170. It is stated that the phase speed predicted is close to the measured $c_M$ values. But this is not the case since $\zeta \sim 1/2$. That suggests $c_M \sim U_T^{1/2}$, which is far far from $\sim U_T^1$. Again, please clarify.

**Reply** *The Eady model can be used to make some quantitative predictions for the experimentally observed baroclinic waves, keeping in mind that the model is highly idealised and therefore some differences can be expected. The differences in the phase speed and also Figure 5 show that simple flow estimations based on the linear theory by Eady are quite limited and cannot easily be carried over to large-amplitude nonlinear waves.*

*We have added this remark to the revised text and corrected a typo: "Differences between the theory and the real flow notwithstanding, the zonal phase speed c predicted by (5) is close ..." there should have been "...by (4) is close...".*

8. **Referee** Section 4: I had trouble connecting the first 3 sections to the analysis in sections 4 and 5. A schematic or two showing how the ARW dynamics lead to these different thermal field properties would be greatly appreciated I believe.

**Reply** *In the first three sections we describe the temperature influence by considering flow features. In section 4 and 5 we focus on statistical properties that depend on the temperature. It is not always possible to directly connect the statistical results with the Rossby wave dynamics. We try this by referring to other work (e.g. Linz et al. (2018) or Dai and Deng (2021)). Moreover, by comparing the findings with the one from reanalysis data in section 5 we hope that our statistical results obtain more reliability. In the introduction of the revised manuscript, when describing the structure of the paper, we point out more clearly that section 4 and 5 focus more on statistics. We think that the reader can then better see the focus of the individual chapters.*

**COMMENTS**

**To:** Referee 2

**From:** Costanza Rodda & Co-authors

**Subject:** Responses to the review

We thank the referee for the valuable comments and efforts towards improving our manuscript. In the following, we comment (italic font) on the referee's suggestions and questions point by point. Modifications are marked in red in the revised manuscript.
* * *
The impact of Arctic amplification resulting from climate change on the atmospheric circulation is an important and topical problem in climate dynamics. This manuscript provides an interesting and unusual approach in exploring the statistics of variability in a plausible laboratory analog of mid-latitude baroclinic circulations, albeit on an f-plane (or at least on a weak topographic beta plane due to free surface curvature). Laboratory studies of fluid dynamical analogs of atmospheric circulation and other phenomena have played an important historical role in helping to validate other approaches to model such phenomena e.g. using numerical tools. It is intriguing, therefore, to see this brave attempt to examine possible insights relevant to climate trends in the Earths atmosphere.

The article itself seems generally well written and the experiments described build on a body of previous work by the Brandenburg/Budapest group. This is clearly just a first step in such studies in that the authors focus on relatively simple statistical measures of changing variability as a function of thermal contrast and latitude (i.e. radius). The results are presented well and discussed realistically, and seem to suggest that the experiments are broadly consistent with recent observations and (at least some) models. I have just a few points of clarification that I would encourage the authors to take into account.

1. **Referee** The authors make reference in several places to the equator in their experiment though such experiments cannot represent realistic equatorial dynamics. I would suggest they replace the term equator with something like subtropics, which is perhaps more accurate dynamically.

   **Reply** *We agree that the experiment does not capture the equatorial dynamics, although the term is often used in the literature. We have replaced the term 'equator' with 'subtropics' in the text when referring to the dynamics.*

2. **Referee** Section 2 notes that the temperature of the outer cylinder is maintained by an electric heater operated at a fixed power input, which one would have thought really corresponds to a fixed heat flux boundary condition rather than a fixed temperature if the same power input is used for all experiments - is that the case? If so, it is surprising from Fig. 2 that merely increasing the temperature of the inner cylinder would lead to a smaller temperature difference across the annulus unless either the power input to the outer bath was reduced or more heat was being lost to the environment instead of being transported by motions in the annulus. This is because a lower temperature difference with fixed heat flux would imply a larger Nusselt number. This deserves a

little more discussion and clarification, since it bears on the degree to which the forcing in the experiment is analogous to Arctic amplification (cf the statement in Lines 127-8).

**Reply** *The cold inner and hot outer baths are obtained differently in our experimental set-up. The water is circulated in the inner cylinder by pipes connecting it to an external device where the water is cooled to a set temperature and then pumped back. The temperature is, therefore, regulated by the thermostat. The outer ring hosts heating wires that heat the water. For all experimental runs, we kept the power supplied to the wires in the outer cylinder fixed and changed the temperature in the cooling thermostat. The system is then let reach the thermal equilibrium before rotation is started. Increasing the temperature in the inner cylinder increases the equilibrium temperature of the working fluid in the middle gap and consequently the equilibrium temperature for the hot bath.*

[Figure]

(a)                                    (b)

Figure 1: Time series of the temperatures measured in the outer (red) and inner (blue) rings for the experiments C11 (a) and C18 (b). The vertical grey line marks when the tank is set into rotation.

*Figure 1 shows the time series of the temperature measured in the inner (in blue) and outer (in red) rings. It can be noticed that when rotation is started (grey vertical line), the temperatures in both baths have reached an equilibrium and remain stable for the entire duration of the experiment. From these considerations, we can conclude that the boundary conditions at the two conductive walls are constant temperature.*

*We have added an appendix with figure 1 and clarified the matter of the boundary conditions.*

3. **Referee** Section 3 describes changes in flow regime with varying thermal contrast at fixed Taylor number (rotation rate), which seems to suggest an increase in dominant wavenumber with thermal Rossby number. This is a little surprising given comparisons with comparable experimental studies by other groups (e.g. see the reviews by Hide & Mason (1975), Read et al. (2014) etc.) where an increase in thermal Rossby number generally leads to lower wavenumber flows. But the Taylor number used here is quite large, placing the flow close to an irregular flow transition which may account for the different behaviour - a point worth mentioning/discussing. Are the results presented here dependent on operating in this high Ta regime?

**Reply** *In the regime identified as 'regular wave' by Hide & Mason (1975) it is indeed the case that an increase in thermal Rossby number leads to lower wavenumber flows. Our experiments have*

*Taylor numbers higher than $10^8$, which is found to be the threshold to irregular baroclinic waves/ geostrophic turbulent regimes for experimental apparatus with similar dimensions to the one used here. In this regime, the flow becomes much more complicated, with irregular structural variation and nonlinear interactions between the dominant mode and sidebands (as discussed in detail by Früh and Read 1997). These complex interactions result in wider spectra spanning several wavenumbers for smaller Ro, as can be seen in figure 4. Although a complete study of the regime transitions and waves interactions is beyond the scope of our paper, it seems that the experiment with the highest $\Delta T$ (C9) is in a regular wave regime, then there is a transition to a structural vacillation with loss of regularity of the waves to finally enter the geostrophic turbulent regime for the smallest $\Delta T$ investigated. We have extended the discussion on the wave regimes, comparing our experiments with other studies in the new manuscript.*

4. **Referee** Minor points:

   - Line 40 'is' rather than 'are'

   **Reply** *Corrected*

   - Line 41 Isn't the key point here that 'cause and effect are difficult to distinguish.' ?

Reply *Yes, thanks for the correction. We updated the text.*

**Referee** - Line 118 The depth of the fluid layer here is fairly shallow, indicating that, although Ta is large the Ekman number (representing the effect of bottom drag) may be relatively large - comment?

**Reply** *This is indeed an interesting point. We dedicated a section in our previous publication [1] to discuss the viscous effects in shallow water configurations. Our previous study reveals that the bottom drag hinders the formation of baroclinic waves for total fluid depth lower than 3 cm (for the same experimental set-up used in the current study). The fluid depth was carefully chosen to avoid the bottom drag affecting the dynamics. We added a paragraph in the manuscript to address this point.*

**Referee** - Line 153 The waves in the experiment are not strictly Eady waves even though the Eady model may have some quantitative comparisons with the experiment. Eady waves are fundamentally baroclinic edge waves associated with thermal gradients along horizontal boundaries (and with weak PV gradients in the interior).

**Reply** *Yes, the baroclinic waves observed in the experiment differ from the highly idealised Eady model. We have modified the text to make this point more clear.*

**Referee** - Line 171 Rossby waves are only slower than the background flow for beta $> 0$. Sloping topography can produce a beta $< 0$ which would lead to faster propagation than the background flow.

**Reply** *Yes, we have added a note to distinguish the two cases in the new text.*

**Referee** - Line 205-6 'the temperature distribution skewness has a positive value at the cold (top) boundary and becomes more and more negative close to the warm (bottom) boundary' Is there a physical interpretation for this?

**Reply** *Yes, the negative skewness is attributed to the passing of a cold thermal front. We added this interpretation to our text.*

**Referee** - Line 262-4 If the outer region between e and the outer cylinder have 'no baroclinic activity', how is heat being transported into the interior?

**Reply** *The heat transport in the baroclinic wave regime is a combination of the transport from the baroclinic eddies and the axisymmetric meridional circulation. In the outer region, which is not reached by the baroclinic eddies, heat transport is mainly due to this meridional circulation. Some minor transport might come from boundary layer instabilities (Th. v. Larcher, S. Viazzo, U. Harlander, M. Vincze, and A. Randriamampianina. Instabilities and small-scale waves within the Stewartson layers of a thermally driven rotating annulus. J. Fluid Mech., 841, 380-407, 2018) and turbulent diffusion.*

**Referee** Line 298 and ff 'not significant' perhaps should be stated as 'not formally significant' since its sign may still be physically significant?

**Reply** *We agree that the trend might be physically significant. We specified it in the manuscript.*

**References**

[1] Rodda, C., Hien, S., Achatz, U. and U. Harlander, (2020) *A new atmospheric-like differentially heated rotating annulus configuration to study gravity wave emission from jets and fronts.* Experiments in Fluids, 61 (2).